# Patient Safety Silence and Safety Nursing Activities: Mediating Effects of Moral Sensitivity

**DOI:** 10.3390/ijerph182111499

**Published:** 2021-11-01

**Authors:** Hyo-eun Jeong, Keum-hee Nam, Heui-yeoung Kim, Yu-jung Son

**Affiliations:** 1Department of Nursing, Kyungnam College of Information & Technology, Busan 47011, Korea; jhe-0324@hanmail.net; 2College of Nursing, Kosin University, Busan 49267, Korea; 3Department of Nursing, Dong-A University Hospital, Busan 49201, Korea; heui-wd@hanmail.net; 4Research Institute of Holistic Nursing Science, College of Nursing, Kosin University, Busan 49267, Korea; sonkalac@hanmail.net

**Keywords:** patient safety, safety management, nurses, hospitals, morals

## Abstract

Among the factors that threaten patient safety and quality of care due to the diversification and complication of hospital environments, nurses play a pivotal role regarding patient safety in the clinical setting. This study investigates the mediating effects of moral sensitivity on the relationship between nurses’ patient safety silence and safety nursing activities and contributes to developing strategies. Nurses (n = 120) employed for at least one year in two university hospitals in Korea between 1 September and 30 October 2020 participated in the study. Data were analyzed using *t*-test, Pearson’s correlation coefficients, and multiple regression using the SPSS/WIN 22.0 program. Additionally, the mediating effects were analyzed using Baron and Kenny’s method and bootstrapping. Safety nursing activities were significantly negatively correlated with patient safety silence and significantly positively correlated with moral sensitivity. Patient safety silence was significantly negatively correlated with moral sensitivity. Moral sensitivity partially mediated the relationship between patient safety silence and safety nursing activities. There is a need to develop and implement individualized ethical programs that enhance moral sensitivity in nurses to promote patient safety nursing activities.

## 1. Introduction

Today, the diversification and complications of hospital systems due to increased patient severity and advances in medical technology are threatening patient safety and quality of care [1]. Safety nursing activities refer to activities that prevent patients’ injuries or accidents that may occur during the delivery of healthcare services [2].

A patient safety incident in the hospital causes physical, mental, and material damage to the patient and caregiver, results in financial loss for the hospital, such as prolonged hospital stay and compensation, and increases the risk for mental repercussions on the involved healthcare provider as the second victim [3]. Thus, patient safety should be considered a responsibility of all members of a hospital organization, and factors threatening patient safety must be identified and managed.

In particular, nurses develop a close relationship with patients as they provide direct care throughout a patient’s hospital stay and are able to detect situations that threaten patient safety [4,5]. Thus, nurses must establish clear goals to improve patient safety and take up an important role in the clinical setting to minimize medical malpractice.

In Korea, there has been growing interest in safety management, which is one of the evaluation criteria for healthcare organization evaluations launched in 2004. With the enforcement of the Patient Safety Act in 2016, patient safety is managed at a national level [6]. Particularly in Korea, medical institutions, patients, and guardians are supposed to voluntarily report patient safety accidents through the Korean patient safety reporting & learning system as part of patient safety management [2,7]. According to the statistics for patient safety incidents, the number of these in Korea has more than tripled over three years, from 3864 in 2017 to 11,953 in 2019, and remains consistently on the rise [7]. Nevertheless, despite the consistent rise in the number of safety incident reports, the reporting of incidents other than those already publicized is not made well [8,9]. Reporting patient safety incidents is crucial, as it aids in identifying errors and system vulnerabilities and leads to the future prevention of patient safety incidents and recurrence [10].

Patient safety silence refers to the choice of remaining silent despite being aware of a practical or potential risk to patients or having, information or ideas to enhance the hospital system in order to improve patient safety [11,12]. Patient safety silence was influenced by individual factors (indifference to or lack of information about patient safety), such as the experience of being exposed to degrading remarks or behaviors by managers when expressing opinions about patient care [13], and structural and environmental factors, for example, a centralized hierarchical structure that emphasizes service, efficiency, and a healthcare culture where individuals do not admit error [14,15]. If nurses remain silent about situations causing patient safety incidents, it would be difficult to detect the risk factors of hospital systems before they cause harm to patients [16]. This can potentially threaten patient safety [17]. Therefore, nurses’ patient safety silence is not an individual’s problem but a major barrier to safety nursing activities and advances in the organization.

Moral sensitivity is the ability to recognize moral conflict, understand patients’ vulnerability situationally and intuitively, and predict ethical outcomes of decisions for patients [18]. Having a high moral sensitivity enables individuals to make responsible decisions in situations including ethical problems [19]. The purpose of nursing activities is to provide quality care based on a moral decision about what is best for the patient. Thus, enhancing moral sensitivity is crucial [20].

Domestic previous studies revealed that Korean nurses tend to avoid raising issues that were identified whilst performing their tasks or items that could lead to improvement in order to avoid deterioration of relationships with colleagues or superiors [11,21]. Furthermore, the nurses who participated in Lee’s study [22] reported that they were reluctant to share problems relating to patient safety incidents. As a consequence, they underreported or remained silent, and were resigned due to a lack of improvement even after offering their opinion. In order to prevent patient safety incidents, it is paramount that nurses working in hospitals refuse to be silent regarding problems relating to patient safety [23].

Moreover, it was revealed that a higher moral sensitivity of nurses correlated with a higher level of compliance with standard precautions, which is a safety nursing protocol [24,25]. As an important concept and a prerequisite for understanding patients’ vulnerable situations and performing ethical work, moral sensitivity was shown to have an influence on decisions relating to nursing behavior [26].

As discussed above, patient safety silence and moral sensitivity are important concepts related to safety nursing protocol, and their influence on each other can be inferred. However, finding studies that comprehensively explored the relationship between them was difficult, and studies examining the mediating effects of moral sensitivity on the relationship between patient safety silence, which is heavily influenced by external factors, and safety nursing activities are even rarer.

Therefore, this study aimed to investigate the mediating effects of moral sensitivity on the relationship between patient safety silence and safety nursing activities among nurses to present foundational data in order to develop strategies promoting safety nursing activities. 

## 2. Materials and Methods

### 2.1. Study Design and Setting

A descriptive cross-sectional study design was employed in this study. Nurses (n = 120) employed for at least one year in two university hospitals in Korea between September 1 and 30 October 2020, participated in the study.

The characteristics of the two university hospitals thus selected were that the number of patient beds was similar at 1000–1100, the ratio of the number of nurses to the number of patients was less than 2:1, and both maintained first grade within Korea´s nursing rating system. Both passed Korea´s Health Care Accreditation, which includes patient safety management evaluation [27,28].

### 2.2. Participants

The inclusion criteria were nurses (n = 120) who had worked in medical and surgical wards, operating rooms, and ICUs that provide direct patient care, such as safety nursing activities. The exclusion criteria were nurses who worked in outpatient clinics, laboratories, and central supply divisions, who were rarely involved in direct patient care. Based on a previous study indicating that newly graduated nurses require 8 months to 1 year to adjust to clinical settings [29], newly graduated nurses with less than one year of work experience were excluded.

The sample size was determined using the G-Power 3.1.9.2 program. For multiple regression analysis at a significance (α) of 0.05, effect size (fz) of 0.15 (medium), power (1-β) of 95%, and 16 predictor variables, the minimum sample size was 107. Considering a 10% dropout rate, data were collected from 120 participants.

### 2.3. Data Collection Instruments

A structured questionnaire was used, and the questionnaire consisted of five items for patient safety silence, 30 items for moral sensitivity, 56 items for safety nursing activities, and 14 items for demographic characteristics, for a total of 105 items.

#### 2.3.1. Patient Safety Silence

This study used the patient safety silence tool for clinical nurses, originally developed by Tangirala and Ramanujam [11] and adapted and validated by Han [23]. The tool comprises five items, and each item is rated on a 5-point Likert scale. Response options included “absolutely not true” (1), “generally not true” (2), “neutral” (3), “generally true” (4), and “absolutely true” (5). Higher scores indicated greater patient safety silence. The reliability of the tool (Cronbach’s α) was 0.82 in the study by Tangirala and Ramanujam [11] and 0.89 in the present study.

#### 2.3.2. Moral Sensitivity

Moral sensitivity was measured using the Moral Sensitivity Questionnaire (MSQ) developed by Lützén et al. [18] and translated into Korean by Han et al. [30] The 30-item MSQ consists of six domains: interpersonal orientation (3), structuring moral meaning (6), benevolence (5), modifying autonomy (3), experiencing moral conflict (5), and trust in medical knowledge and principles of care (7), with one unclassified item. The average rating of each item was used to calculate the MSQ score. Scores ranged from 1 to 7, where a higher score indicated higher moral sensitivity. The reliability of the tool (Cronbach’s α) was 0.78 in the study by Lützén et al. [18] and 0.83 in the present study.

#### 2.3.3. Safety Nursing Activities

This study used the safety nursing activities tool modified and adapted by Jang [31], that was developed by Cho [32] for ICU nurses for use by ward nurses. The tool was developed based on the patient safety criterion of hospital accreditation evaluation criteria of the Korea Institute for Healthcare Accreditation [7]. The tool used by Jang [31] consists of 56 items across ten factors: medication administration (9), blood transfusion (9), patient care during transport (8), infection (12), patient identification (5), communication (4), pain (2), pressure ulcers (2), falls (2), and environment (3). Each item is rated on a 5-point Likert scale ranging from never to always (1–5). A higher score indicated greater safety in nursing activities. The reliability of the tool (Cronbach’s ⍺) was 0.97 in the study by Jang [31] and 0.98 in the present study.

### 2.4. Ethical Considerations

Data were collected after obtaining approval from the Institutional Review Board at Kosin University (No. KU IRB 2020-0038). In adherence to the Declaration of Helsinki, the purpose, anticipated benefits, and potential risks and inconveniences of the study were explained to the directors of nursing, head teaching nurses of the two university hospitals, and all participants. Participants were required to fill an informed consent form (including information regarding anonymity, confidentiality, voluntary consent, and freedom to decline participation) before the study. They completed the questionnaire in an isolated space, not interfered with by the head nurse or nursing managers, to ensure confidentiality. Participants were told they could withdraw from the study at any time. The author’s contact information was specified on the consent form so that the participants could contact the author if they had any questions.

The collected data were stored in a designated place and managed by the principal investigator to prevent data leakage. The collected data were processed on a computer and stored in a portable hard drive designated by the principal investigator. The computer containing the collected data was password-protected to allow access only to the researchers. All matters pertaining to security were managed by the principal investigator. The study data will be safely discarded using a shredding machine 3 years after the report, and the files stored in the study computer and data in the portable drive will be permanently deleted.

### 2.5. Data Analysis

The collected data were analyzed using IBM SPSS Statistics for Windows version 22.0. (IBM Corp: Armonk, NY, USA). General characteristics were analyzed with real numbers and percentages. The level of patient safety silence, moral sensitivity, and safety nursing activities were analyzed as mean and standard deviation. The correlations between patient safety silence, moral sensitivity, and safety nursing activities were analyzed using Pearson’s correlation coefficients. The hypothesis that patient safety silence affects safety nursing activities through the mediation of moral sensitivity was tested using the three steps of Baron and Kenny’s [33] method. The significance of the mediating effects of moral sensitivity (mediator) on the relationship between patient safety silence and safety nursing activities was tested using the PROCESS macro version 3.3.(Model No. 4) of SPSS by Hayes [34], bootstrapping with 5000-resamples was performed to test the statistical significance of the indirect effect.

## 3. Results

### 3.1. Participants’ General Characteristics

Table 1 shows the general characteristics of the participants. In terms of the nurses’ age composition, the first reference point was set based on the reported results that the average number of working years of Korean nurses was six years, and nurses´ average age was 28 years [35,36]. In addition, the second reference point was set based on the reported results that the older the nurses´ age, the more the moral maturity or wisdom accumulated, and the higher the moral sensitivity when they were old than 40 years of age [26]. Half of the participants (50%) were between the ages of 28–39 years, with a mean age of 33.88 ± 9.408 years. The majority of the participants were female (92.5%; n = 111), and 36.7% of the participants (n = 44) had a clinical career of one to four years. The majority (99.2%; n = 119) of the participants had prior patient safety education, and 75% (n = 90) had been involved in patient safety incidents.

### 3.2. Level of Patient Safety Silence, Moral Sensitivity, and Safety Nursing Activities

The mean patient safety silence score was 2.080 ± 0.748 out of 5, and the mean moral sensitivity score was 4.911 ± 0.486 out of 7. The mean safety nursing activity score was 4.489 ± 0.459 out of 5 (Table 2).

### 3.3. Correlations between Patient Safety Silence, Moral Sensitivity, and Safety Nursing Activities

Safety nursing activities had a significant negative correlation with patient safety silence (r = −0.337, *p* < 0.001) and significant positive correlation with moral sensitivity (r = 0.359, *p* < 0.001). Patient safety silence was significantly negatively correlated with moral sensitivity (r = −0.248, *p* = 0.006) (Table 3).

### 3.4. Mediating Effects of Patient Safety Silence on the Relationship between Moral Sensitivity and Safety Nursing Activities

Table 4 and Figure 1 show the results of the mediation analysis for moral sensitivity in the relationship between patient safety silence and safety nursing activities.

Multiple regression analysis was performed to analyze the mediating effects of moral sensitivity. Before the analysis, we checked whether the assumptions of the regression analysis were satisfied. The Durbin Watson index for autocorrelation was above 0.1, close to 2, with a range of 1.564–1.673, confirming independence; the variation inflation factor (VIF) was below 10, at 1.066, thus, confirming the absence of multicollinearity among the independent variables (Table 4).

The results showed that moral sensitivity mediates the effect of patient safety silence on safety nursing activities (Figure 1, Table 4). In step 1, patient safety silence (independent variable) significantly influenced moral sensitivity (mediator) (β = −0.248, *p* = 0.006), with an explained variance of 5.4%.

In step 2, patient safety silence (independent variable) significantly influenced safety nursing activities (dependent variable) (β = −0.337, *p* < 0.001), with an explained variance of 10.6%. In step 3, patient safety silence and moral sensitivity were entered as independent variables, and safety nursing activities were entered as the dependent variables. Both patient safety silence (β = −0.264, *p* = 0.003) and moral sensitivity (β = 293, *p* = 0.001) significantly affected safety nursing activities. In other words, with moral sensitivity as the mediator in step 3, patient safety silence had a significant effect on safety nursing activities, with the regression coefficient (β) decreasing to −0.264 from −0.337 in step 2. This confirms that moral sensitivity has a partial mediation. These variables explained 18.1% of the variance in safety nursing activities. The upper and lower limits of the 95% confidence interval were −0.0952–−0.0089 and did not include 0, confirming statistical significance (Table 4).

## 4. Discussion

This study examined the levels and relationships between patient safety silence, moral sensitivity, and safety nursing activities among hospital nurses and verified the mediating effect of moral sensitivity on the relationship between patient safety silence and safety nursing activities.

First, we examined hospital nurses’ safety nursing activities based on our findings. The mean safety nursing activity score, the dependent variable, was 4.49 out of 5, which is relatively high. Previous studies reported a mean score of 4.05 out of 5 among 188 hospital nurses in Korea [37] and 4.18% out of 5 among 175 operation room nurses in Korea [33], showing that Korean nurses engage in a high level of safety nursing activities.

This is speculated to be due to the continued education and training which is used to practically promote patient safety and quality of care based on systematic guidelines for safety nursing activities, due to the implementation of the healthcare organization accreditation system in Korea. However, we cannot generalize this result, as most studies compared were on nurses of general hospitals with experience with healthcare organization accreditation. Moreover, rather than treating safety nursing activities, which is essential to fostering a safe hospital environment, as a part of preparing for healthcare organization accreditation, healthcare organizations should recognize the importance of safety nursing activities even if they do not undergo healthcare organization accreditation and develop a system and strategies for continuous patient safety management and quality improvement.

Second, we examined the relationships between patient safety silence, moral sensitivity, and safety nursing activities based on our results. Our results showed that safety nursing activities increased with decreasing patient safety silence. In the study by Doo and Kim [21] on the impact on patient safety in nurses of general hospitals with 500 beds or more, greater organizational silence had a negative impact on patient safety. Considering that organizational silence and patient safety silence are similar constructs based on past findings that organizational and structural factors influence patient safety silence, this result supports our findings. Nurses’ patient safety silence begins from indifference to patient safety and leads to intentional and strategic avoidance of reporting patient safety incidents due to lack of knowledge and information about patient safety incident reporting, fear of punishment, and concerns with the outcome [23]. Additionally, being silent about situations related to patient safety incidents hinders the detection of risk factors in hospital systems [38], ultimately affecting patient safety [17]. In particular, based on the results of Schwappach and Gehring [39], where patient safety silence persists when errors such as failure to confirm prescription and medication preparation are repeated, hospital managers should pay attention and provide support to increase nurses’ awareness of patient safety by ensuring nurses’ personal experiences and interests, education within each nursing unit, learning organizations, and quality improvement activities are continued.

Furthermore, moral sensitivity has a positive effect on safety nursing activities. These results were similar to the results of the study by Han et al. [24] on moral sensitivity and compliance with infection standard precautions in 214 general hospital nurses in Korea, where compliance with infection-related standard precautions significantly increased with increasing moral sensitivity. This suggests that patient safety-related behavior is associated with nurses’ internal responsibility and sense of ethics. Thus, customized strategies to develop human resources are needed to improve moral sensitivity and promote quality safety nursing activities.

Finally, we examined the mediating effects of moral sensitivity on the relationship between patient safety silence and safety nursing activities. In our study, moral sensitivity was confirmed to have a significant partial mediating effect on the relationship between nurses’ patient safety silence and safety nursing activities. This result suggests that safety nursing activities may differ even among nurses with similar tendencies regarding patient safety silence based on their individual moral sensitivity.

Moral behaviors are shown through moral problem-solving when moral consensus is reached through moral sensitivity. Hence, moral sensitivity, which refers to the ability to recognize moral problems and understand the impact of one’s behavior on others, is a key factor in the process of moral decision-making [40]. Moral sensitivity is particularly important in attenuating patients’ pain because nurses emotionally respond to patients’ suffering and recognize and interpret the moral meaning, value, and duties of a situation through their moral sensitivity, which leads to moral behaviors [41].

Although we could not directly compare our findings to the literature due to the rarity of studies that examined the mediating effects of moral sensitivity on the relationship between patient safety silence and safety nursing activities, this study is significant as it confirmed the partial mediation of moral sensitivity on patient safety silence and safety nursing activities. More replication studies should be conducted to confirm the mediating effects of moral sensitivity in improving safety nursing activities.

Nurses’ moral sensitivity can be enhanced through training and education, and based on the findings of a Korean study that moral sensitivity was higher with more experience with moral education [42], nursing ethics programs should first be developed and implemented when devising strategies to promote safety nursing activities in nurses. Because the degree of moral sensitivity differs across individuals, and moral sensitivity cannot be altered with one-time education, individualized education protocols and simulation-based programs to strengthen and integrate moral judgment should be developed to provide continuous and systematic education and management.

The limitations of this study are as follows. A large-scale randomized study is suggested in the future, since the extracted sample cannot accurately represent the population, as the study was conducted via the convenience sampling method targeting nurses in certain university hospitals. Furthermore, as this study is a cross-sectional correlation study that cannot reveal the causal relationship between variables, a longitudinal study is needed in the future to affirm the causal relationship between variables.

Nonetheless, in order to reduce the negative effects of nurses´ patient safety silence on safety nursing activities, the mediating effect of moral sensitivity, which was not attempted in previous studies, was confirmed. Therefore, this study is significant in that it provides data on the basis of interventions that can improve safety nursing activities.

## 5. Conclusions

This study is a descriptive survey to examine the relationship between nurses’ patient safety silence and safety nursing activities. It also investigates the mediating effects of moral sensitivity. Safety nursing activities were significantly negatively correlated with patient safety silence and significantly positively correlated with moral sensitivity. Patient safety silence was significantly negatively correlated with moral sensitivity. Additionally, moral sensitivity partially mediated the relationship between patient safety silence and safety nursing activities.

This study proves the necessity of moral sensitivity in the relationship between patient safety silence and safety nursing activities and will serve as important basic data for developing nursing strategies that can improve safety nursing activities.

## Figures and Tables

**Figure 1 ijerph-18-11499-f001:**
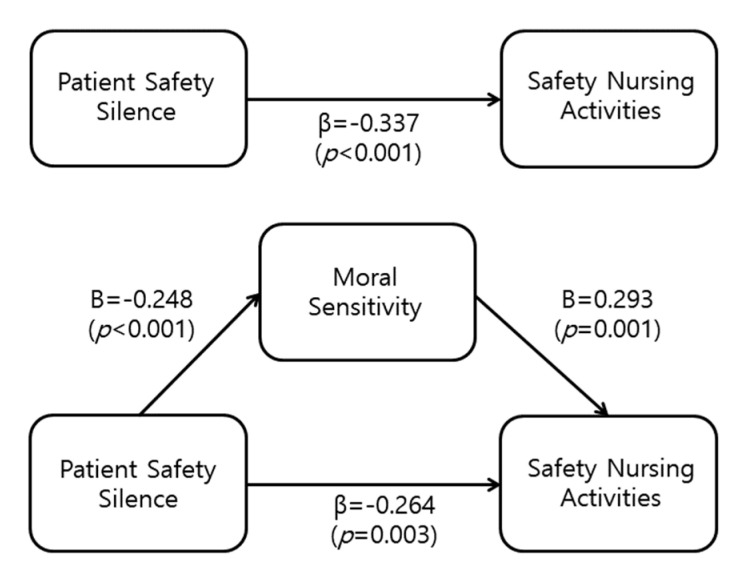
Mediating effects of moral sensitivity on the relationship between patient safety silence and safety nursing activities (n = 120).

**Table 1 ijerph-18-11499-t001:** General characteristics of participants (n = 120).

Variables	Categories	n	%	Mean (SD)
Age (years)	<28	30	25	33.88 (9.408)
	28∼<40	60	50	
	≥40	30	25	
Sex	Female	111	92.5	
	Male	9	7.5	
Career (years)	<1	9	7.5	8.538 (7.838)
	1~<5	44	36.7	
	5~<10	26	21.7	
	≥10	41	34.2	
Education	Diploma degree	19	15.8	
	Bachelor’s degree	95	79.2	
	≥Master’s degree	6	5.0	
Working unit	Medical ward	16	13.3	
	Surgical ward	60	50.0	
	Intensive care unit	12	10.0	
	etc.	32	26.7	
Role	Staff nurse	79	65.8	
	Charge nurse	27	22.5	
	Nurse manager	11	9.2	
	No answer	3	2.5	
Participation of patient safety education	Yes	119	99.2	
	No	1	0.8	
Experiences of the patient safety incidents	Yes	90	75.0	
	No	30	25.0	

**Table 2 ijerph-18-11499-t002:** The levels of patient safety silence, moral sensitivity, safety nursing activities (n = 120).

Variables	M ± SD	Range
Patient Safety Silence	2.080 ± 0.748	1∼5
Moral Sensitivity	4.911 ± 0.486	1∼7
Safety Nursing Activities	4.489 ± 0.459	1∼5

**Table 3 ijerph-18-11499-t003:** Correlation of patient safety silence, moral sensitivity, and safety nursing activities (n = 120).

	Patient Safety Silence	Moral Sensitivity	Safety Nursing Activities
	r (P)	r (P)	r (P)
Patient Safety Silence	1		
Moral Sensitivity	−0.248 (0.006)	1	
Safety Nursing Activities	−0.337 (<0.001)	0.359 (<0.001)	1

**Table 4 ijerph-18-11499-t004:** Mediating effects of moral sensitivity on the relationship between patient safety silence and safety nursing activities (n = 120).

Causal Steps				B	β	Adj *R^2^*	F (t)	*p*
Step 1.	Patient Safety Silence	→	Moral Sensitivity	−0.161	−0.248	0.054	7.747	0.006
Step 2.	Patient Safety Silence	→	Safety Nursing Activities	−0.207	−0.337	0.106	15.142	<0.001
Step 3.	Patient Safety Silence & Moral Sensitivity	→	Safety Nursing Activities			0.181	14.119	<0.001
(1)	Patient Safety Silence	→	Safety Nursing Activities	−0.162	−0.264		−3.087	0.003
(2)	Moral Sensitivity	→	Safety Nursing Activities	0.277	0.293		3.423	0.001

## Data Availability

All data generated or analyzed during this study are included in this published article.

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
