# Peer review of "Patient Safety Silence and Safety Nursing Activities: Mediating Effects of Moral Sensitivity"

_ijerph, 2021, doi:10.3390/ijerph182111499_

Round 1

Reviewer 1 Report

Dear Authors,

I have the following comments regarding the article:

  1. In the Introduction section, it would be necessary to comment on the considerable increase in patient safety incidents in Korea (lines 31-32), and also to provide more information about patient safety system in Korea (lines: 49-52). Moreover, it is advisable to describe what results have been obtained in Korea in the discussed subject matter so far, and then to justify undertaking the research and the rationale of the study.
  2. In Materials and Methods: the para 'Study design and settings' should be better described, instead of repeating the objective of the study once again.

It is also important to include information on the two university hospitals, describe their characteristics, the population size being subjected to the health care services and resources (e.g. number of beds, number of nurses employed there, and proportion of the nursing staff in these hospitals in relation to the general number of hospital nurses in Korea), and finally, the structure/organization of patient safety system.

Line 109: please specify the names of the three questionnaires used in this study.

Line 162: real numbers should be replaced by number of observations.

  1. Results section; lines 198-200 and 214-215: there is an information about methodology which could be scheduled and moved to the Methods section. Further, 'bootstraping' should be better elucidated.
  2. In the Discussion, please attempt to describe limitations and strenghts of the study.
  3. In the final section 'Conclusions', I would recommend rephrasing the text. Please delete the sentence in lines 296-302, that contain duplicated information from the Results. The conclusions should be kind of a response to the purpose of the study.
  4. Please, highlight the changes to the revised version using a different color and/or a different script.

Author Response

Thank you very much for your careful review and for another opportunity to submit the revised manuscript. We appreciate the helpful comments and believe that the paper has been greatly improved.

We have revised the manuscript and highlighted the revisions in red.

Reviewer 2 Report

The reviewer's opinion is that: 

1) Consider whether the topic should add that it is about specific research. The given topic has a general dimension.

2) Regardless of the comments made in the discussion, write in the conclusions about the limitations that result from the place of the conducted research.

(3) It would be good to indicate the perspective of further research. 

4) A minor remark. Namely, in line 57 there is an unnecessary sign "0". 

Author Response

(The authors gave the same response as above.)

Reviewer 3 Report

This paper tries to highlight the impact of patient safety silence on nurse patient safety activities and the relation between these elements and the moral sensitivity. This is an interesting and meaningful issue and minor revisions are required before considering the publication:

  • In the introduction section could be useful to give a definition of patient safety culture.
  • In ‘Data analysis’ paragraph should be made explicit what kind of test do the SPSS macro PROCESS.
  • Authors should explain why or how they choose the three age categories.
  • Line from 198 to 200 should be moved to the method section.
  • Pag 7 line 244: the word ‘silent’ is missing in ‘with decreasing patient safety [silent]’.
  • The authors should list the limits of the study.

Author Response

(The authors gave the same response as above.)
